# Spaceborne Relative Radiometer: Instrument Design and Pre-Flight Test

**Duo Wu** [1,2], **Wei Fang** [1,*], **Kai Wang** [1], **Xin Ye** [1], **Ruidong Jia** [1], **Dongjun Yang** [1], **Baoqi Song** [1], **Zhitao Luo** [1], **Yuwei Wang** [1], **Zhiwei Xia** [1], **Ping Zhu** [3,*] and **Michel van Ruymbeke** [4]

1    Changchun Institute of Optics, Fine Mechanics and Physics, Chinese Academy of Sciences, Changchun 130033, China
2    School of Optoelectronics, University of Chinese Academy of Sciences, Beijing 100049, China
3    Institute for Advanced Study, Shenzhen University, Shenzhen 518060, China
4    Royal Observatory of Belgium, 1180 Brussels, Belgium
*    Correspondence: fangw@ciomp.ac.cn (W.F.); pzhu@szu.edu.cn (P.Z.)

**Abstract:** In order to simultaneously determine the values of total solar irradiance (TSI) and the Earth's radiation at the top of the atmosphere (TOA) on board the Fengyun-3F satellite, a spaceborne relative radiometer (SRR) was developed. It adopts a dual-channel structure, including a solar radiometer channel (SR) with an unobstructed field of view (FOV) of 1.5° and an Earth radiometer channel (ER) with a wide field of view (WFOV) of 95.3° and a diameter of about 1900 km on the ground. Before the launch, both the SR and ER were calibrated. The SR, installed on the inner frame of the solar tracker of the SIM-II (solar irradiance monitor-II), is used to observe rapid changes in solar radiance with the SIAR (solar irradiance absolute radiometer), an electrical-substitution radiometer, on orbit. The ER is mounted on the U-shaped frame of the solar tracker, directly pointing in the nadir direction. Additionally, a dark space observation mode is used to determine the on-orbit background noise and lunar observation mode for on-orbit calibration. In this article, the instrument design and working principle of the SRR is first introduced, and an analysis of the measurement model of the ER, the WFOV channel of the SRR, is focused on. Finally, ground test results of the SRR are introduced.

**Keywords:** relative radiometer; Earth radiation budget; on-orbit calibration; measurement model

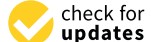



## 1. Introduction

The Earth's climate system is powered by solar radiation. The temporal and spatial differences in the distribution of solar radiation at the Earth's surface provide energy for surface water evaporation, convection, precipitation, wind, and ocean circulation, which contribute to the diversity of the Earth's climate [1–6]. Meanwhile, some of the incoming solar radiation is reflected by the atmosphere and the Earth's surface, which is known as reflected shortwave radiation (RSR). Additionally, the Earth also emits long-wave thermal radiation, outgoing long-wave radiation (OLR), into space. This kind of energy flow that the Earth receives from the Sun and gives back to outer space is defined as Earth's radiation budget (ERB), which is one of the 54 essential climate variables (ECVs) determined by the global climate observing system (GCOS) [7]. In a steady Earth's climate system, the incoming and outgoing radiation at the TOA stays in a dynamic balance. However, anthropogenic forcing is usually considered to break this balance and cause continual Earth energy imbalance (EEI). Researchers analyzed Earth radiation data from satellites and Argo and found that the EEI value doubled during the period from 2005 to 2019 [8,9]. Scientists from the Intergovernmental Panel on Climate Change (IPCC) also warn that global warming of 1.5 K will be reached between 2030 and 2050 at the current rate [10]. Therefore, the measurement of the ERB is very important to study climate change and global warming.

The measurement of the ERB mainly consists of two parts. One of them is the TSI measurement. The other is the measurement of terrestrial outgoing radiation (TOR). The TSI value has been continuously measured from space since the 1970s. There are six sets of instruments in operation to measure the TSI from space, as shown in Figure 1.

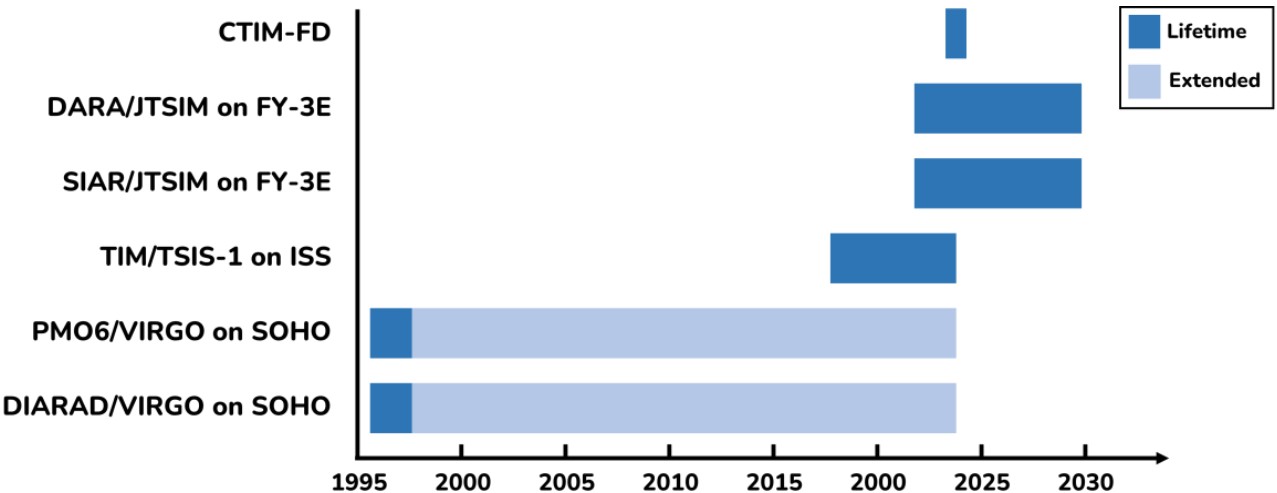

**Figure 1.** Instruments in operation measuring TSI from space. DIARAD: Differential Absolute Radiometer; VIRGO: Variability of Irradiance and Gravity Oscillations; SOHO: Solar and Heliospheric Observatory; PMO: Physikalisches und Meteorologisches Observatorium; TIM: Total Irradiance Monitor; TSIS: Total and Spectral Solar Irradiance Sensor; ISS: International Space Station; SIAR: Solar Irradiance Absolute Radiometer; JTSIM: Joint Total Solar Irradiance Monitor; FY: Feng Yun; DARA: Digital Absolute Radiometer; CTIM-FD: Compact Total Irradiance Monitor Flight Demonstration.

The VIRGO experiment on the SOHO mission has been measuring the TSI value on orbit for more than 27 years, covering two complete solar cycles. TIM/TSIS-1 on ISS continues the on-orbit TSI value measured by a TIM-type absolute radiometer, since 2003. The JTSIM-DARA experiment on the FY-3E mission continues an on-orbit TSI value measured by a SIAR-type radiometer and a DARA-type absolute radiometer. This is also the third time that two independent absolute radiometers have been used to measure the TSI value on the same mission. The CTIM-FD is a radiometer verifying new technology used for TSI measurement. In addition to the instruments shown in Figure 1, the Space Cryogenic Absolute Radiometer (SCAR) is also being considered for TSI observation [11–16].

The observation of the TOR, including outgoing longwave radiation, reflected short-wave radiation, and total radiation, at the TOA, started in 1975 with the Nimbus-6 satellite. Figure 2 shows a number of representative instruments in operation measuring the TOR from space. There are six CERES instruments in operation to provide the shortwave, longwave, and total radiance products with different temporal resolutions. The ERM instrument on FY-3C measures the Earth's radiance in shortwave and total bands on a Sun-synchronous orbit. GERB4 observes the Earth's radiance in shortwave and total bands from a geostationary orbit [17–20].

On the FY-3F satellite, we developed a spaceborne relative radiometer to monitor the total solar irradiance and observe the terrestrial outgoing radiation in 0.2–20.0 μm. This was achieved via two different channels of the SRR: the SR and the ER. The SR is a narrow field of view (NFOV) channel installed on the inner frame of the solar tracker, which will measure the TSI value and calibrate with the SIAR. The ER is a WFOV broadband channel mounted on the U-shaped frame of the solar tracker pointing in the nadir direction. The ER will observe the total radiance reflected and emitted from the Earth's disk at about 1900 km in diameter. The SRR shares a similar measurement principle with the bolometric oscillation sensor (BOS) that was on the PICARD mission from 2010 to 2014 [20,21]. Unlike the 180° FOV of the BOS, the updated NFOV design of the SR channel and WFOV of the ER

channel allow the SRR to have enhanced stray light suppression ability during TSI and TOR observations. The SR channel uses the dark space mode to calibrate the background noise and compare it with the on-orbit TSI measurement results of the SIAR. The ER channel uses the dark space mode and lunar observation mode to determine the background noise and perform on-orbit calibration. Compared with the previous wide field of view ERB measurements, the SR and ER use the identical bolometer type detector, and the two are only different in terms of optical design. This configuration can minimize the differences in the instruments themselves caused by the measurement of the TSI and EOR by two different instruments.

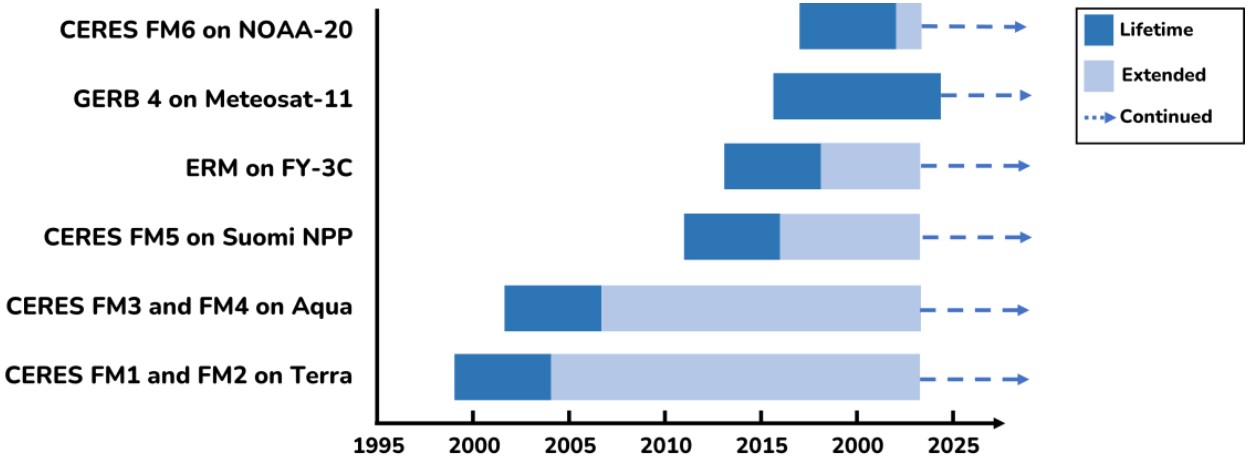

**Figure 2.** Instruments in operation measuring Earth's outgoing radiation (EOR) from space. CERES: Clouds and the Earth's radiant energy system; FM: Flight module; NPP: National polar-orbiting partnership; ERM: Earth radiation budget; GERB: Geostationary Earth radiation budget; NOAA: National oceanic and atmospheric administration.

This article first presents the instrument design of the SRR, including the optical design, operation mode, and on-orbit calibration method. In order to adapt to the optical design, the measurement model based on the bolometer principle is analyzed and established. Benefiting from the operation mode of the SR, the absolute value of the SR measurement is traced to the SI unit by comparing the total solar irradiance measurement in the same measurement period with the electrical-substitution absolute radiometer SIAR. The calibration of the ER measurement is achieved through the dark space and lunar observation mode. Section 2 provides an overview of the SRR instrument design. The measuring model of the SRR based on the bolometer principle is analyzed in Section 3; Section 4 describes the pre-flight tests of the SR and ER. Some findings are discussed in Section 5; Section 6 provides the conclusions.

## 2. Instrument Design

### 2.1. Overview

Figure 3 shows the schematic diagram of the on-orbit observation of the SRR on board the FY-3F satellite. The SRR/FY-3F consists of two 0.2–20 μm total wavelength channels; the SR channel has a 1.5° NFOV for the TSI measurement, and the ER channel has a 95.3° WFOV for Earth radiation observation at an altitude of about 836 km. Each channel contains a planar thermal detector based on the bolometer principle and an optical system. In the SR channel optical system, there is a reflector applied in front of the thermal detector to increase the absorptivity of the planar thermal detector. A baffle with an extinction thread was designed for the ER optical system to reduce the influence of solar stray light during observation. The measurement results of the SR and ER were sampled using electronic systems.

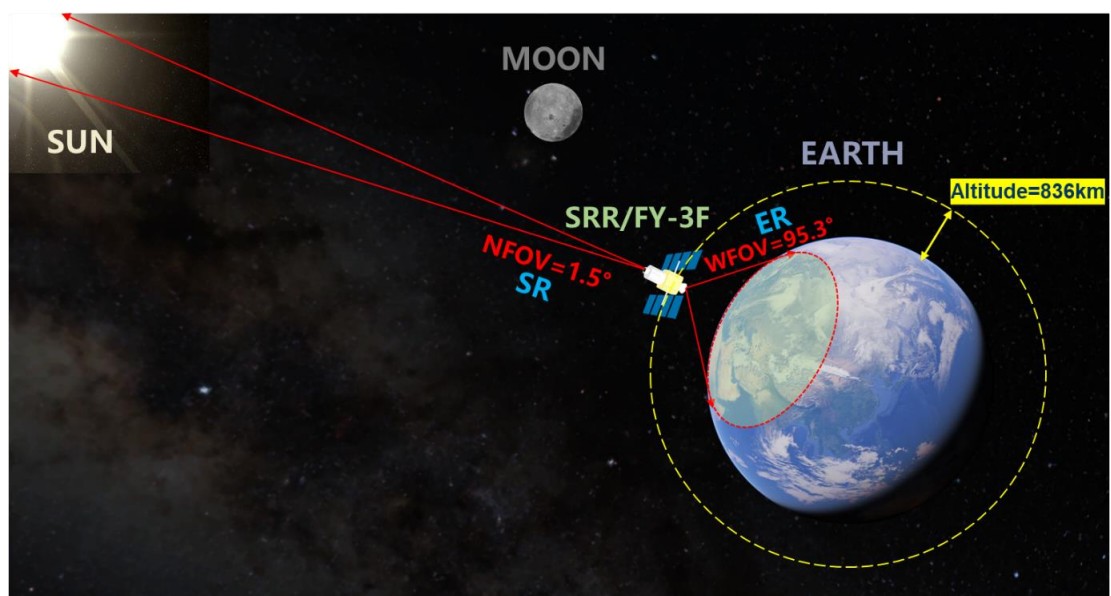

**Figure 3.** Schematic diagram of SRR space observation on FY-3F satellite in Sun-synchronous orbit.

As shown in Figure 4a, to track the Sun during the TSI measurement, the SR is installed on the solar tracker of the SIM-II/FY-3F, which points to the Sun by using a Sun sensor for feedback during the TSI measurement mode on each orbit. The ER channel is mounted on the U-shaped frame of the solar tracker. It is designed to point in the nadir direction during the whole mission, except in dark space mode and lunar observation mode. The pointing accuracy of the ER is guaranteed by the satellite platform. The control unit of the SIM-II/FY-3F is shown in Figure 4b.

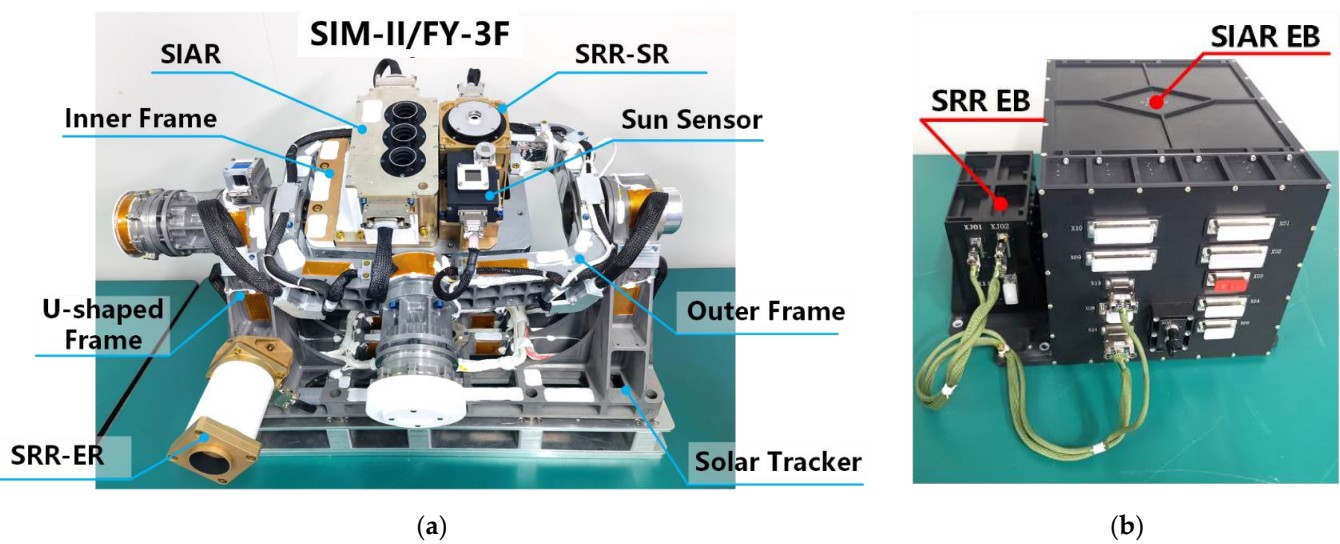

<div align="center">(<b>a</b>)     (<b>b</b>)</div>

**Figure 4.** Photographic image of SIM-II instrument and electronic box: (**a**) Sensor box and solar tracker of SIM-II/FY-3F; (**b**) Electronic box of SRR and SIAR.

## 2.2. Sensitivity Analysis of the Design

The SR and ER of the SRR use the planar thermal detector based on the bolometer principle. The planar thermal detector consists of a heat rod and two COTS thermistors, as shown in Figure 5. The detector is in an axial thermal conductivity structure configuration. The temperature distribution of the heating rod can represent the change in the power of the incident light absorbed by the black coating.

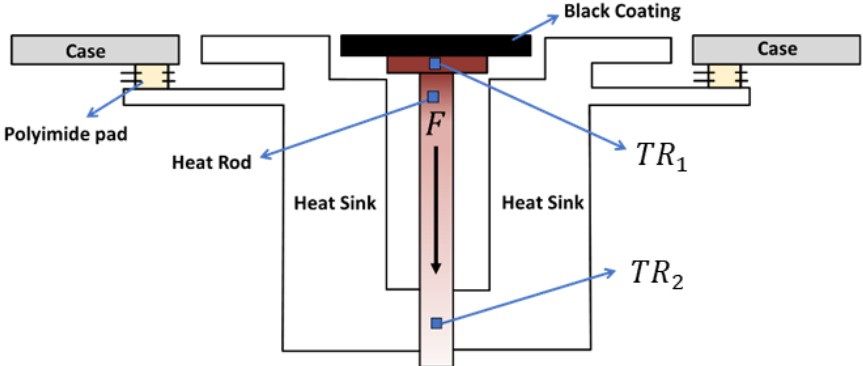

**Figure 5.** Schematic diagram of planar thermal detector.

The planar thermal detector is painted with high-absorptance and high-emissivity black coatings. The absorptivity is 96.6% and the emissivity is 91% over the whole spectrum range. The thickness of the black coating is about 32 µm. Additionally, $T_1$ and $T_2$ are measured with the 100K6MBD NTC thermistors working in a temperature range of 233.5 K to 398.5 K. High-thermal-resistance Polyimide pads are used to suppress the thermal interference from the case of the detector.

Because the amplitude of the EEI is generally below 1 W/m², requirements are put forward for the measurement sensitivity of the instrument, and the measurement sensitivity should be above 0.1 W/m². The minimum energy received by the detector $\Phi_{min}$ can be calculated by Equation (1)

$$\Phi_{min} = S \times A_{SR} \tag{1}$$

The $\Phi_{min}$ is about 7.54 µW. The thermal connection structure needs to be designed so that when the incident radiation power is 7.54 µW, the temperature gradient between the receiving surface of the detector and the heat sink at a steady state should be greater than 0.1 mK. Therefore, the thermal conductivity G of the thermal detector is designed to be:

$$G = \frac{\Phi_{min}}{\Delta T} = 0.0754 W/K \tag{2}$$

The heat conductivity should be less than 0.0754 W/K, and at the radiated power input of 7.54 µW, there will be a temperature gradient greater than 0.1 mK. The specific design parameters are shown in Table 1, after calculation by Equation (3)

$$G_{design} = k\frac{A}{l} = 0.0712 W/K \tag{3}$$

**Table 1.** The parameter of planar detector of SRR/FY-3F.

| Parameter | Value |
| --- | --- |
| Thermal conductivity, k | 238 W/m/K |
| Length of heat rod, l | 94.5 mm |
| Diameter of heat rod, d | 6 mm |

The design can meet the sensitivity requirements. In addition, according to the temperature characteristics of the thermistor, when the temperature changes by 1 K near 25 °C, the resistance of the thermistor changes by about 4%. It can be simply deduced that when the temperature changes by 1 mK, the thermistor change should be 0.004%. For thermistor-type radiation detectors, in order to meet the relative measurement error, it is usually necessary to have the ability to detect temperature changes of 0.1 mK. Taking the 100K6MBD small thermistor produced by the TE company used in the planar radiation detector of the space relative radiometer as an example, when the temperature changes

to 0.1 mK near 298.15 K, the resistance of the 100K6MBD thermistor changes to 0.0004%, which is converted to a resistance change of 0.4 Ω. Assuming that the excitation voltage across the resistor is 5 V, when the temperature changes by 0.1 mK, the thermistor changes by 0.4 Ω, and the output voltage changes by about 5 μV at this time. In order to perform μV-level voltage measurement, the noise from the circuit during signal amplification first needs to be suppressed.

### 2.3. Optical Design

Compared with the cavity-type thermal detector, the incident light that hits the planar detector cannot undergo enough reflections for absorption. A parabolic reflector was designed to solve this problem. This method was first used in the CTIM-FD instrument, which also uses planar detectors for TSI measurement. Figure 6a shows the optical structure of the SR channel adopted from [15].

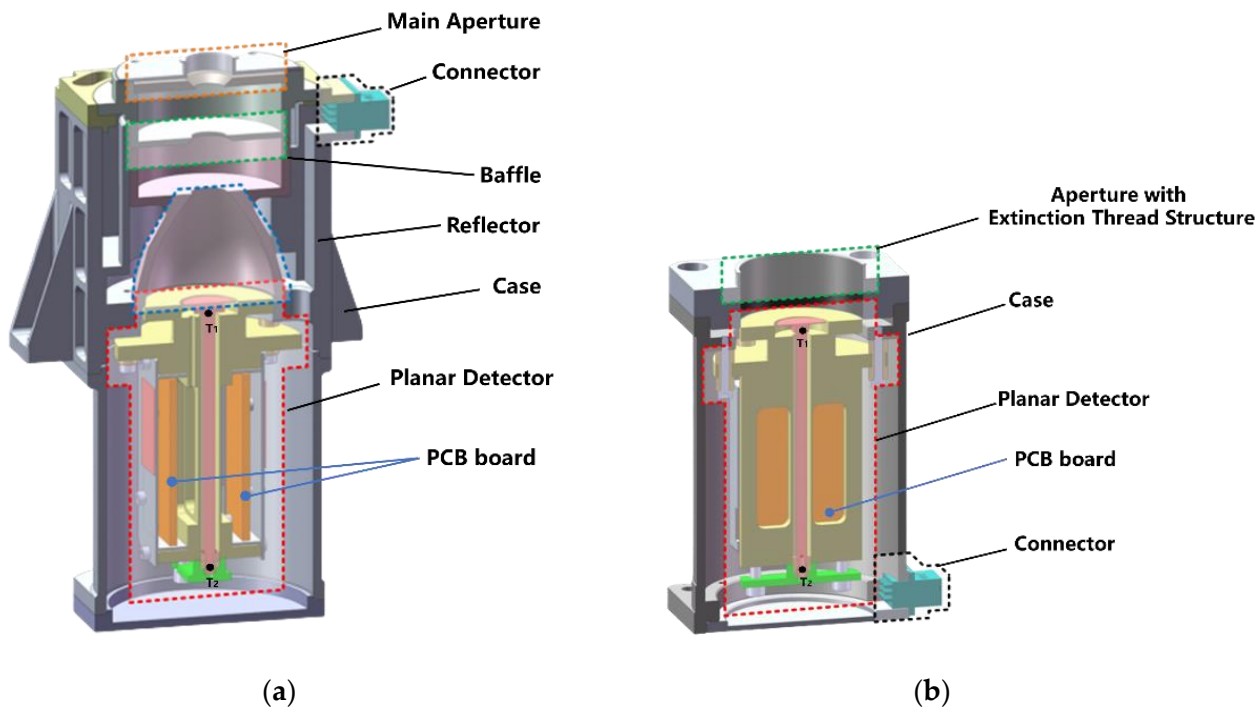

**(a)**                                                                                     **(b)**

**Figure 6.** Mechanic model of the SRR: (**a**) Optical structure of SR; (**b**) Optical structure of ER.

The main aperture is to limit the entrance area of the incident light. The baffle is painted with black coatings to suppress stray light. The parabolic reflector was designed using Equation (4):

$$y^2 = 2px \tag{4}$$

where the focal parameter $p = 45$. The center of the black coating area of the SRR detector is placed on the focus of the parabolic reflector, as shown in Figure 7. The entrance of the parabolic reflector is 12 mm in diameter and absorption area of the detector is 18 mm in diameter. When an incident light ray is reflected from the focus, it will return to the focus point again after two reflections by the parabolic reflector. The parabolic reflector will increase the absorption of the planar detector by creating reflection paths to capture the first reflected light rays.

Unlike the NFOV design of the optical system of the SR channel, the wide FOV of the ER, which allows the observation of more radiation from Earth, is valued more during the design of the optical system of the ER channel. A black-coated extinction thread, shown in Figure 6b, is used to suppress stray light from the Sun.

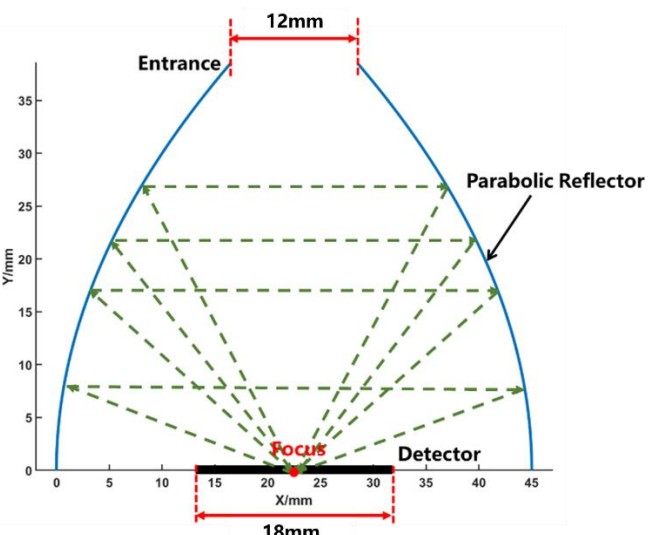

**Figure 7.** Schematics diagram of the parabolic reflector used in SR.

### 2.4. Electronic System

The function block diagram of the electronic system of the SRR is shown in Figure 8. Pre-AMP circuit boards are mounted on the thermal detectors of the SR and ER, as illustrated in Figure 6a,b. In the Pre-AMP circuit, a lock-in technique with a square wave is applied. The dual D-type Flip-Flop provides the modulation signal for the resistor bridge and the demodulation signal for the analog switch. During this process, the circuit noise is suppressed.

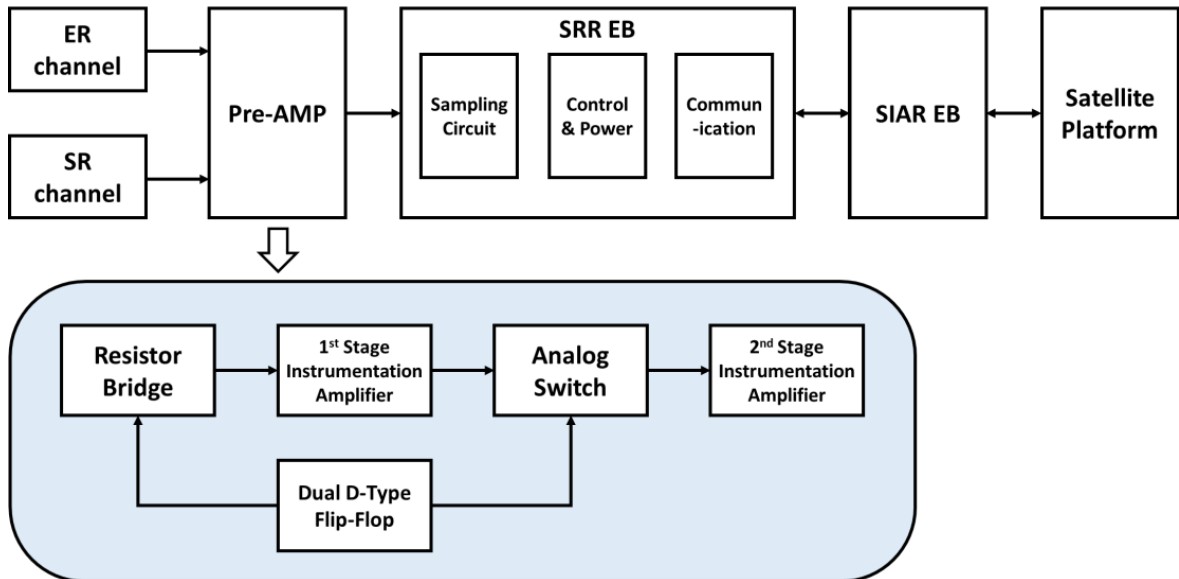

**Figure 8.** Function block diagram of the electronic system of SRR.

The SRR EB sends measurement data to the SIAR EB every second via an RS422 interface. The SIAR EB then packs SRR measurement level 0 data together with a timestamp, orbit information, the case temperature of the SRR, and other required information into a data package and sends it to the satellite platform every minute. In addition, the case temperature of the ER and SR channels is controlled by the SIAR EB to achieve a stable environment during the operation. The SIAR EB also provides the system power for the SRR instrument, making it possible to restart the SRR to reduce the risk due to software failure.

## 3. Working Principle and Measurement Model

### 3.1. Working Principle

　　As mentioned in Section 1, the measurement principle is based on the thermal transfer theory [22]. Figure 9 is a diagram of the energy budget on the thermal detector with a 180° FOV. This can be described through Equation (5), as follows:

$$P_{\text{incident}} = P_{\text{internal}} + P_{\text{emit}} + P_{\text{flux}} \tag{5}$$

where $P_{\text{incident}}$ is the power of the incident light; $P_{\text{emit}}$ is the power emitted by the black-coated surface of the detector; $P_{\text{internal}}$ is the internal power of the detector; and $P_{\text{flux}}$ is the power of thermal conduction via the heat rod.

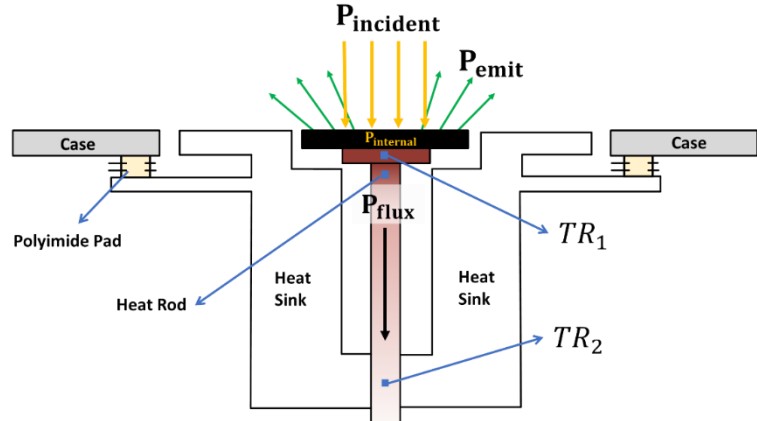

**Figure 9.** Schematic diagram of the energy budget of the thermal detector with 180° FOV.

### 3.2. Measurement Model of Detector with 180° FOV

　　As shown in Figure 10, the radiation network method was applied to analyze the detector measurement model in TSI measurement. $J_1$ is the radiosity of the detector surface. $J_2$ is the radiosity of the Sun. $J_3$ is the radiosity of the dark space. Shape factor $F_{1,2}$ is the fraction of energy leaving the detector surface that reaches the solar surface. Shape factor $F_{1,3}$ is the fraction of energy leaving the detector surface that reaches the dark space.

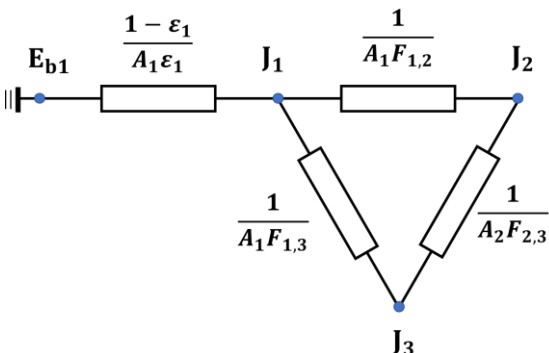

**Figure 10.** Radiation network for detector with 180° FOV.

　　According to the radiation network analysis, when we apply Kirchhoff's current law to the network, it states that the sum of heat current flows into the node $J_1$ is zero, as described in Equations (6) and (7):

$$\frac{E_{b1} - J_1}{\frac{1-\varepsilon_1}{A_1\varepsilon_1}} + \frac{J_2 - J_1}{\frac{1}{A_1F_{1,2}}} + \frac{J_3 - J_1}{\frac{1}{A_1F_{1,3}}} = 0 \tag{6}$$

$$-\frac{E_{b1} - J_1}{\frac{1-\varepsilon_1}{A_1\varepsilon_1}} = P_{\text{internal}} + P_{\text{flux}} \tag{7}$$

where $E_{b1}$ is the emissive power of the detector surface; $A_1$ is the black-coated area of the detector surface; and $\varepsilon_1$ is the emissivity of the detector surface. It assumes that the emissivity is equal to the absorptivity of the detector surface. The solar energy that reaches the detector surface is written as

$$J_2 A_1 F_{1,2} + J_3 A_1 F_{1,3} = P_{\text{internal}} + P_{\text{flux}} + J_1 A_1 \tag{8}$$

Solving $J_1 A_1$ from Equation (8) and making use of Equation (7) gives

$$\varepsilon_1 J_2 A_1 F_{1,2} + \varepsilon_1 J_3 A_1 F_{1,3} = P_{\text{internal}} + P_{\text{flux}} + P_{\text{emit}} \tag{9}$$

where $\varepsilon_1 J_2 A_1 F_{1,2}$ is contributed by solar irradiance. Additionally, $\varepsilon_1 J_3 A_1 F_{1,3}$ is contributed by the dark space radiation. According to the Stefan–Boltzmann law, $P_{\text{emit}}$ can be expressed as Equation (10):

$$P_{\text{emit}} = A_1 \varepsilon_1 E_{b1} = A_1 \varepsilon_1 \sigma T_1{}^4 \tag{10}$$

where $\varepsilon_1$ is the emissivity of the black-coated area of the detector; $\sigma$ is the Stefan–Boltzmann constant; and $T_1$ is the temperature measured using the thermistor $TR_1$. In this case, the background is the dark space 2.7 K temperature. Therefore, the background radiation is not considered in the equation. The $P_{\text{flux}}$ can be described from Equation (11):

$$P_{\text{flux}} = G_{\text{rod}}(T_1 - T_2) \tag{11}$$

where $G_{\text{rod}}$ is the thermal conduction of the heat rod; $T_2$ is the temperature measured using the thermistor $TR_2$. $P_{\text{internal}}$ can be calculated as described in Equation (12):

$$P_{\text{internal}} = Cm_1 \frac{\partial T_1}{\partial t} \tag{12}$$

where $C$ is the thermal capacity of the detector. $m_1$ is the mass of the detector. $\frac{\partial T_1}{\partial t}$ is the temperature change at the $TR_1$ point. The $P_{\text{incident}}$ can then be stated as follows:

$$P_{\text{incident}} = \varepsilon_1 J_2 A_1 F_{1,2} + \varepsilon_1 J_3 A_1 F_{1,3} = A_1 \varepsilon_1 \sigma T_1{}^4 + G_{\text{rod}}(T_1 - T_2) + Cm_1 \frac{\partial T_1}{\partial t} \tag{13}$$

assuming that the dark space temperature is zero. So,

$$P_{\text{incident}} = \varepsilon_1 J_2 A_1 F_{1,2} = A_1 \varepsilon_1 \sigma T_1{}^4 + G_{\text{rod}}(T_1 - T_2) + Cm_1 \frac{\partial T_1}{\partial t} \tag{14}$$

and with $T_1$ and $T_2$, Equation (14) provides a measurement model to determine the solar energy that reaches the detector's surface during the TSI measurement in space.

### 3.3. Measurement Model of Detector with Limited FOV

The main limitation of the measurement model presented in Section 3.2 is that it is only suitable for a relative radiometer with a 180° FOV. Because the FOV of the SR and ER channel of the SRR are limited by optical systems, the temperature of the optical system needs to be taken into account.

Figure 11 shows the radiation network for a detector with a NFOV or a WFOV. $J_4$ is the radiosity of the aperture surface. In the SR, $J_2$ is the radiosity of the solar surface. In the ER, $J_2$ is the radiosity of the Earth's surface. $F_{1,4}$ is the fraction of energy leaving the detector surface that reaches the inner surface of the aperture. $F_{1,3}$ is the fraction of energy leaving the detector surface of the aperture that reaches the dark space. $F_{1,2}$ is the fraction of energy leaving the detector surface that reaches the solar, or the Earth's, surface.

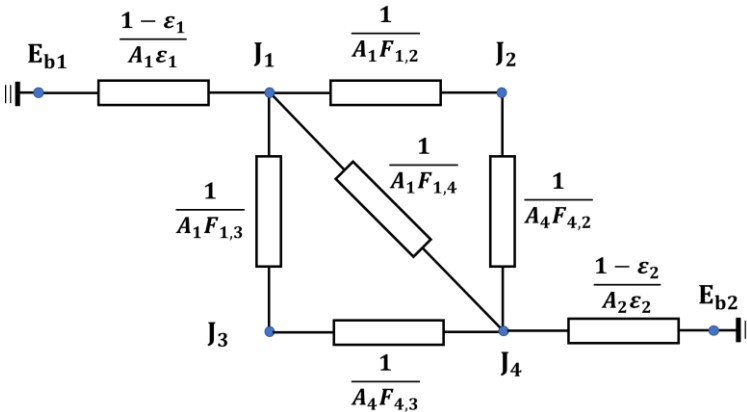

**Figure 11.** Radiation network for detectors with NFOV or WFOV.

Kirchhoff's current law can be applied into node $J_1$, so that

$$\frac{E_{b1} - J_1}{\frac{1-\varepsilon_1}{A_1\varepsilon_1}} + \frac{J_2 - J_1}{\frac{1}{A_1 F_{1,2}}} + \frac{J_3 - J_1}{\frac{1}{A_1 F_{1,3}}} + \frac{J_4 - J_1}{\frac{1}{A_1 F_{1,4}}} = 0 \tag{15}$$

$$-\frac{E_{b1} - J_1}{\frac{1-\varepsilon_1}{A_1\varepsilon_1}} = P_{\text{internal}} + P_{\text{flux}} \tag{16}$$

The total power received by the detector can be calculated from Equations (15) and (16).

$$\varepsilon_1 J_2 A_1 F_{1,2} + \varepsilon_1 J_3 A_1 F_{1,3} + \varepsilon_1 J_4 A_1 F_{1,4} = P_{\text{internal}} + P_{\text{flux}} + P_{\text{emit}} \tag{17}$$

Assuming that the dark space temperature is zero and $J_4 = E_{b2}$, Equation (17) can be written as:

$$\varepsilon_1 J_2 A_1 F_{1,2} = P_{\text{internal}} + P_{\text{flux}} + P_{\text{emit}} - \varepsilon_1 E_{b2} A_1 F_{1,4} \tag{18}$$

Equation (18) can be changed into

$$\varepsilon_1 J_2 A_1 F_{1,2} = A_1 \varepsilon_1 \sigma T_1^{\,4} + G_{\text{rod}}(T_1 - T_2) + Cm_1 \frac{\partial T_1}{\partial t} - \varepsilon_1 \varepsilon_2 \sigma T_{opt}^{\,4} A_1 F_{1,4} \tag{19}$$

where $T_{opt}$ is the temperature of the inner surface of the optical system; and $\varepsilon_2$ is the emissivity of the inner surface of the optical system. With the measured $T_1$, $T_2$, and $T_{opt}$ values and the calculated shape factors $F_{1,2}$ and $F_{1,4}$ (In the ER condition, the $F_{1,2}$ is about 0.5205. $F_{1,4}$ is about 0.4982) [23,24], Equation (19) gives the measurement model of a detector with a NFOV and a WFOV.

## 4. Pre-Flight Tests of SRR

In order to verify the performance of the instrument before launch, a ground comparison traced to the World Radiometric Reference (WRR) for the SR and a vacuum extended area blackbody test for the ER channel were carried out.

Figure 12 is a photo of the comparison test of the SR channel. The SR channel was installed on the solar tracker together with the SIAR instrument and SIAR-2C absolute radiometer, which has been used in the IPC (International Pyrheliometer Comparisons). A shutter was applied before the main aperture of the SR channel in order to obtain the zero point and remove the background basis of the measurement.

The comparison test provides the evidence to evaluate the measurement consistency between SR and SIAR-type radiometers. Unlike the electrical-substitution radiometer, the SR channel cannot obtain the on-orbit WRR calibration factor in this test. This may be explained by the influence of convection. Compared with detectors in the vacuum, convection provides additional thermal paths for detectors in SATP (standard ambient

temperature and pressure) conditions, which may decrease the thermal response of the detectors. In order to suppress the influence of convection, a linear correction method based on the measurement model mentioned in Section 3 was applied.

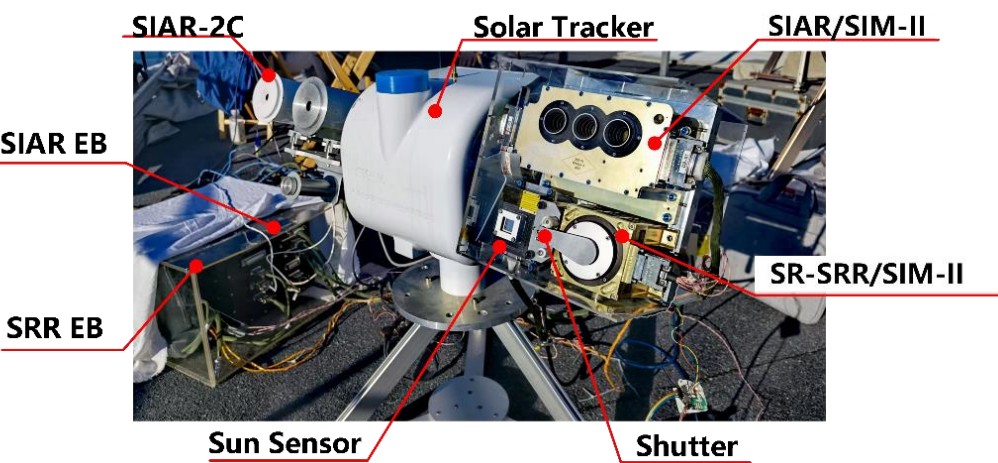

**Figure 12.** Photographic image of the comparison test of SR.

Figure 13 shows the results of a comparison between the SR and SIAR-2C. The measurement results of SIAR-2C are shown in green diamonds. The measurement period of SIAR-2C was 2 min. In order to compare them with results from SIAR-2C, the SR measurement results with a measurement period of 1 s were corrected. The blue line shows the measurement results of the SR with a 1 s period. Additionally, the red circles are the results extracted from the SR results every 2 min.

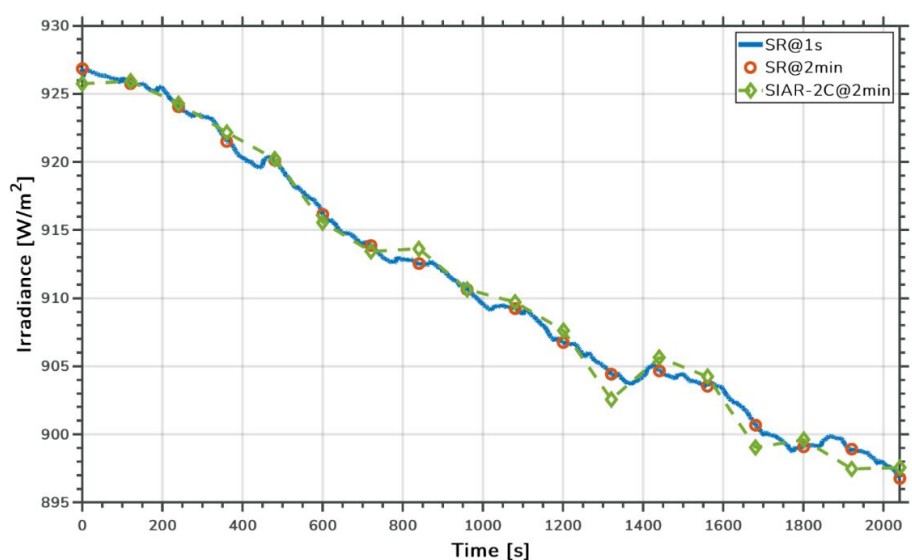

**Figure 13.** Results of comparison test between SR and SIAR-2C.

The Pearson correlation coefficient between results of the SR within a 2 min period and SIAR-2C is about 0.99545. There is a strong correlation between measurement results of the SR and SIAR-2C. Additionally, the SR could also provide second-scale solar irradiance data during the 2 min period.

The aim of the ER is to measure the TOR at the TOA with a spectrum range from 0.2 to 20 $\mu$m. A lack of independent SW and LW channels will limit the ER in further providing LOR and RSR products. However, the Sun-synchronous orbit with ECT 10:00 may offer a chance for the ER to obtain the TOR when observing the Sun shadow area.

The Pearson correlation analysis was applied to the measurement results, as shown in Figure 14. The Pearson correlation coefficient is about 0.99545.

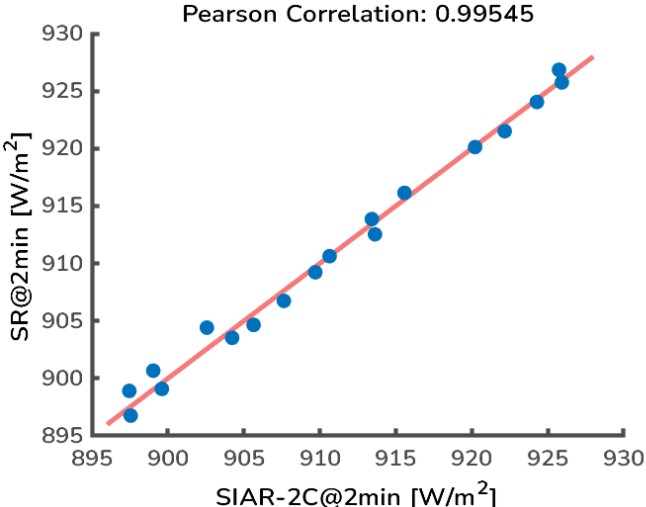

**Figure 14.** Pearson correlation analysis of SR and SIAR-2C measurement results.

A test of the performance of the ER was carried out in a vacuum chamber by using an extended-area blackbody developed by the NIM (National Institute of Metrology, China), as shown in Figure 15. According to the test report of the blackbody, the emissivity of the blackbody is 0.9902, and the temperature homogeneity is better than 0.06 K. The platform in the vacuum chamber was cooled to 233 K in order to provide a cooling path for the L-shaped support and the ER channel. The extended-area blackbody was controlled by a DD-200F refrigerated circulator. A PRT installed in the blackbody was used to measure the temperature and control the feedback of the blackbody. The thermistor provided the temperature profiles of the support during the test.

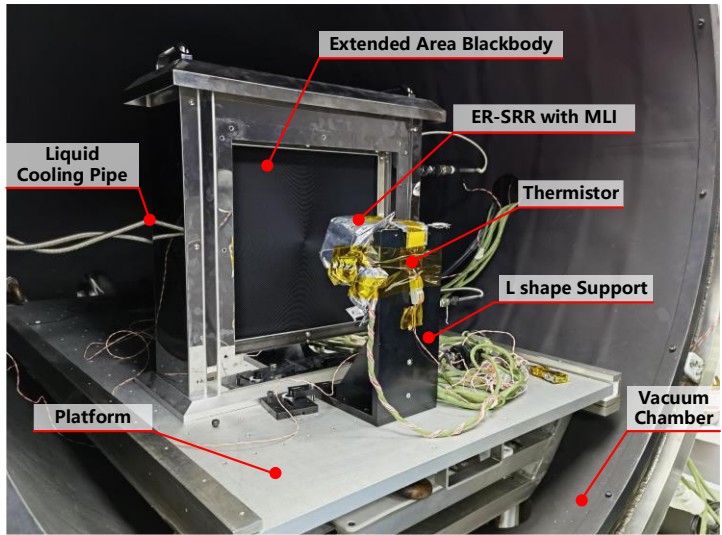

**Figure 15.** Photographic image of the performance test of ER.

The case temperature of the ER was controlled to around 273.15 K during the test. Additionally, the blackbody temperature changed from 258.15 K to 283.15 K in 5 K increments. Figure 16 displays the change in temperature and the irradiance of the blackbody during the test. By applying the measurement model to this test, the blackbody irradiance was $J_2$

in Equation (16). Additionally, it could be calculated using the measurement results of $T_1$, $T_2$, and $T_{opt}$ and the shape factors $F_{1,4}$ and $F_{1,2}$.

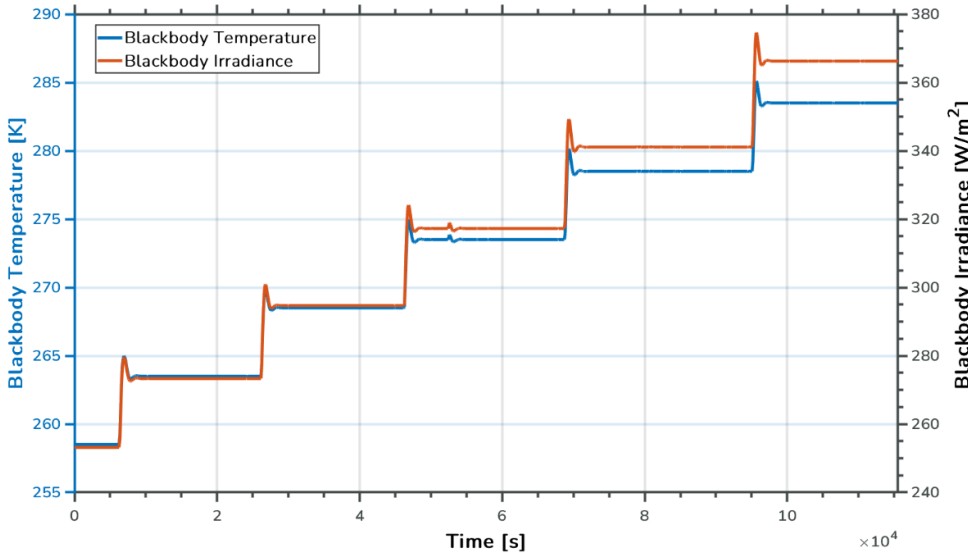

**Figure 16.** The temperature and irradiance of the blackbody during the test.

The shape factors $F_{1,4}$ and $F_{1,2}$ could be calculated. Therefore, the relation between the standard irradiance from the blackbody and the irradiance calculated from the measurement results is provided in Figure 17.

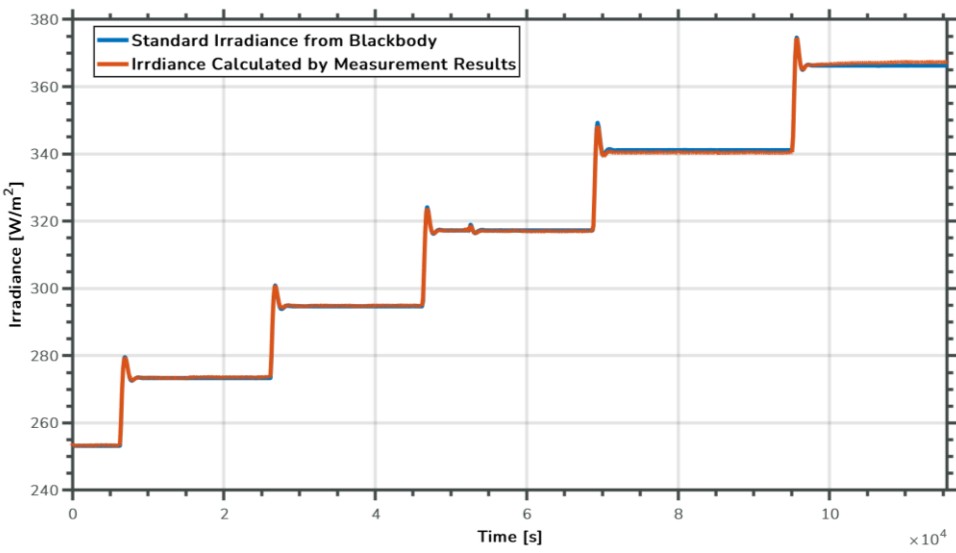

**Figure 17.** The relation between standard irradiance from blackbody and that calculated using measurement results.

Figure 18 shows the error between the standard irradiance and the calculated irradiance. As we can see in this figure, the error increased during the blackbody temperature adjustment period. Additionally, the error varied when the blackbody was at a different temperature. The peak-to-peak error value was approximately 2.1 W/m$^2$.

The measurement results of the ER can provide an estimate for the standard irradiance within 0.7% based on the measurement model mentioned in Section 3.

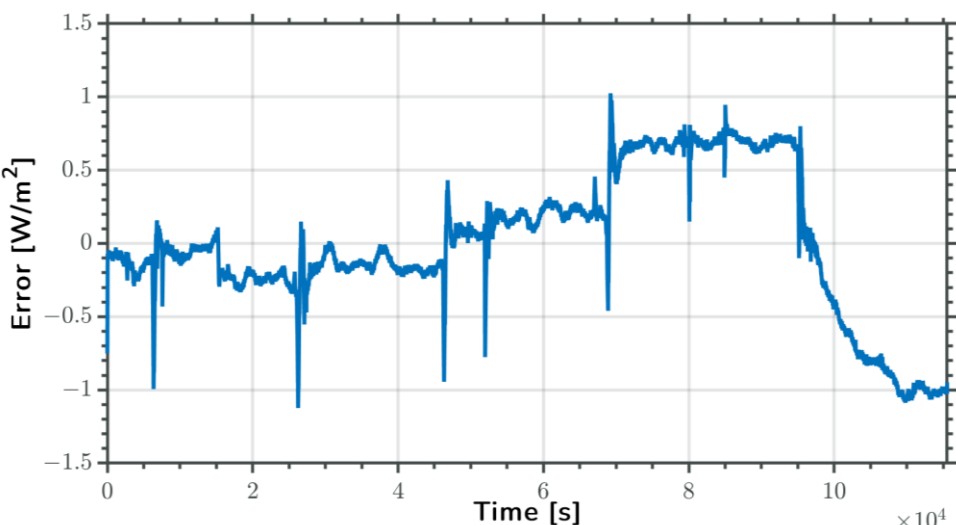

**Figure 18.** Errors between standard irradiance from blackbody and irradiance calculated using measurement results.

## 5. Discussion

A relative radiometer based on the bolometer principle with the aim to measure the ERB at the TOA was developed. The SR channel and ER channel with different optical systems can achieve a TSI and TOR measurement in the spectral range of 0.2–20 μm, respectively.

There are some issues to consider in terms of the instrument design. The absorptivity of the planar radiation detector needs to be enhanced. To increase the absorptivity, a polished parabolic reflector was designed for the SR channel, providing additional reflection paths to capture incident light rays. Different from the SR channel, solar stray light is the main problem in the ER channel. Considering the WFOV and instrument dimension, a black-coated extinction thread is used to suppress stray light. The analysis of the measurement model is another important aspect. By analyzing the contribution of thermal radiation with the radiation network method in a typical working case, the measurement model of the limited FOV relative radiometer was built. The shape factors and the temperature of the optical system were taken into consideration.

To verify the performance of the SRR instrument, pre-flight tests were conducted with the SR and ER channels. With the rapid measurement mode, the SR can provide TSI results with a resolution of 1 s between the measurement period of the SIAR absolute radiometer. Additionally, a Pearson correlation coefficient of about 0.99545 between the SR and SIAR-2C was achieved in the comparison test infield campaign. We noticed the difference between the SR and SIAR-2C. This may be due to the different convection effects received by different instruments. The SR and SIAR-2C simultaneously sense the TSI and SR has a 1-s cadence; however, SIAR-2C has a 120-s sampling rate. Furthermore, the SR is installed side by side with three cavities of electrical substitute absolute radiometers, which allow an on-orbit cross-calibration of the SR channel's absolute level in terms of irradiance. This idea has been verified on a ground-based solar irradiance measurement campaign. The SR has an excellent agreement with the absolute radiometer SIAR-2C within 0.3%, which indicates the uncertainty level SR of FY3-F could be one order of magnitude lower than the previous PICARD-BOS measurements. The results of some previous studies indicate that the sensitivity and time constant of a radiometer will increase in a vacuum due to the absence of a convection component in a heat transfer path. The on-orbit calibration of the SR will be realized by comparing TSI values in the same period between the SR and SIAR absolute radiometer. During the pre-flight test, a standard extended blackbody source was applied to evaluate the performance of the ER in a vacuum. The irradiance measurement error of the ER was within 0.7% of the standard blackbody radiation value

when the blackbody temperature increased from 258.15 K to 283.15 K. Although COTS thermistors were used in this design to ensure the consistency of the responses of different thermistors, the differences in the temperature response between $TR_1$ and $TR_2$ may still have caused the fluctuation in errors in different blackbody temperatures. Thermistors need to be carefully calibrated in order to obtain improved measurement performances. Dark space measurements need to be performed in both the SR and ER channels to determine the zero point. The ER can also use the lunar observation mode for in-flight calibration.

**6. Conclusions**

The SR and ER make up part of the SIM-II, which aims to measure the total solar irradiance onboard the Fengyun meteorological satellite. TSI and solar spectral irradiance (SSI) are two fundamental solar parameters that are continuously monitored with the solar package of Fengyun satellites. The absolute level of TSI is determined at the 0.1% level among different absolute radiometers. The SSI has a much larger discrepancy due to the influence of nonlinear degradation in the short wavelength range [25]. The planar detector has the advantage of enabling a rapid response to electromagnetic radiation at the second level compared to a cavity-based absolute radiometer with a much more compact volume and light weight. In terms of science return, the SR and ER observation will provide a solution to monitoring the global average radiation flux, including seasonal and interannual changes, and will eventually be used in the study of the Earth's energy imbalance.

**Author Contributions:** The work was realized with the collaboration of all the authors. Conceptualization, W.F., P.Z. and X.Y.; methodology, D.W., W.F., X.Y., P.Z. and M.v.R.; software, D.W., D.Y. and B.S.; validation, D.W., W.F., P.Z. and X.Y.; formal analysis, D.W., P.Z. and Z.X.; investigation, D.W., W.F., P.Z., X.Y., K.W., R.J., Z.X., Z.L., Y.W., D.Y. and B.S.; resources, W.F., P.Z. and X.Y.; data curation, D.W.; writing—original draft preparation, D.W., W.F. and P.Z.; writing—review and editing, D.W., W.F. and P.Z.; visualization, D.W.; supervision, W.F., P.Z. and X.Y.; project administration, W.F., P.Z. and X.Y. All authors have read and agreed to the published version of the manuscript.

**Funding:** This research was funded by the National Science Foundation of China (Grant Number: 41974207).

**Data Availability Statement:** The data presented in this study are available on request from the corresponding author.

**Acknowledgments:** This work was partly supported by the International Fund Program of Changchun Institute of Optics, Fine Mechanics and Physics, Chinese Academy of Sciences, the International Space Science Institute (ISSI) in Bern, through ISSI International Team project #500 (Towards Determining the Earth Energy Imbalance from Space) and the Moonbase exploration research equipment purchase project of the Development and Reform Commission of Shenzhen Municipality (No. 2106-440300-04-03-901272).

**Conflicts of Interest:** The authors declare no conflict of interest.

**Nomenclature**

| | |
|---|---|
| AMP | Amplifier |
| BOS | Bolometric Oscillation Sensor |
| CERES | Clouds and the Earth's Radiant Energy System |
| COTS | Commercial Off-The-Shelf |
| CTIM-FD | Compact Total Irradiance Monitor Flight Demonstration |
| DARA | Digital Absolute Radiometer |
| DIARAD | Differential Absolute Radiometer |
| EB | Electronic Box |
| ECT | Equatorial Crossing Time |
| ECVs | Essential Climate Variables |
| EEI | Earth Energy Imbalance |
| EOR | Earth's Outgoing Radiation |
| ER | Earth Radiometer Channel |

| ERM | Earth Radiation Measurement |
| FM | Flight Module |
| FOV | Field of View |
| FY | Feng Yun |
| GCOS | Global Climate Observing System |
| GERB | Geostationary Earth Radiation Budget |
| IPCC | Intergovernmental Panel on Climate Change |
| ISS | International Space Station |
| JTSIM | Joint Total Solar Irradiance Monitor |
| LW | Long-Wave |
| NFOV | Narrow Field of View |
| NIM | National Institute of Metrology, China |
| NOAA | National Oceanic and Atmospheric Administration |
| NPP | National Polar-orbiting Partnership |
| NTC | Negative Temperature Coefficient |
| OLR | Outgoing Long-wave Radiation |
| PMO | Physikalisches und Meteorologisches Observatorium |
| PRT | Platinum Resistance Thermometers |
| RSR | Reflected Shortwave Radiation |
| SATP | Standard Ambient Temperature and Pressure |
| SIAR | Solar Irradiance Absolute Radiometer |
| SIM-II | Solar Irradiance Monitor-II |
| SOHO | Solar and Heliospheric Observatory |
| SR | Solar Radiometer Channel |
| SRR | Spaceborne Relative Radiometer |
| SW | Short-Wave |
| TIM | Total Irradiance Monitor |
| TOA | Top of the Atmosphere |
| TOR | Terrestrial Outgoing Radiation |
| TSI | Total Solar Irradiance |
| TSIS | Total and Spectral Solar Irradiance Sensor |
| VIRGO | Variability of Irradiance and Gravity Oscillations |
| WFOV | Wide Field of View |
| WRR | World Radiometric Reference |

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
