# Peer review of "Spaceborne Relative Radiometer: Instrument Design and Pre-Flight Test"

_remotesensing, doi:10.3390/rs15123085_

Round 1

Reviewer 1 Report

This manuscript reflects the complete design, development and testing of the instrument, and has high readability and engineering reference. But the way it is currently written makes it not look like an academic paper, and it is clearly very different from the journal's paper style. The manuscript's current state is more like a working report, without presenting and arguing for clear research arguments and innovations. It is recommended that the author make changes, such as adding an analysis part of the design. What were the challenges of making such an instrument? How the key technical indicators are broken down, what innovative methods are employed, and so on.

Author Response

Dear Reviewer,

Thank you for your insightful comments on our manuscript. We appreciate your positive feedback regarding the design, development, and testing of the instrument, as well as its readability and engineering reference value.

Point 1: The way it is currently written makes it not look like an academic paper, and it is clearly very different from the journal's paper style. The manuscripts current state is more like a working report, without presenting and arguing for clear research arguments and innovations. It is recommended that the author make changes, such as adding an analysis part of the design. What were the challenges of making such an instrument? How the key technical indicators are broken down, what innovative methods are employed, and so on.

Response 1: We acknowledge your concern regarding the current writing style of the manuscript. We apologize for any confusion. To address this issue, we add a sensitivity analysis of the design section that discusses the challenges encountered during the instrument's development. About the key technical indicator, we have broken down the sensitivity into the thermal parameters of the detector and requirement of the electronic system. The innovative methods are the thermal design of the detector and the lock-in Pre-Amp and high-resolution V/F sampling electronic system. This will ensure that the research arguments and innovations are clearly presented and well-supported.

The original intention of this article is to give an overall introduction to the SRR/FY-3F instrument, hoping that readers can have a good understanding of the instrument. We have also planned another article to introduce the instrument design in detail. We thank you for bringing this to our attention, and we will make the appropriate revisions to improve the manuscript accordingly. Your feedback is highly valuable, and we appreciate your guidance throughout this process.

If you have any further suggestions or comments, please feel free to let us know.

Best regards,

Duo Wu

Reviewer 2 Report

Dear all,

The manuscript aims to reveal and introduce the calibrations of the instruments for recording the values of Total Solar Irradiance (TSI) and Earth’s radiation at the Top of the Atmosphere (TOA) onboard the Fengyun-3F satellite before it is launched. This is to obtain high-quality data and accurate data. The result reveals that thermistors and dark space measurements inside instruments are needed to be carefully calibrated and performed to overcome the rate of errors.

Comments will be provided below: -

1- You could clearly mention the novelty of your work in a sentence in the introduction based on previous studies.

2- You could also add extra results of the study in a couple of sentences for clarity of (what is exactly you reached to in the study).  

Best regards,

Author Response

Dear Reviewer,

Thank you for your valuable comments and suggestions on our manuscript. We appreciate your feedback and have addressed your points accordingly.

Point 1:  You could clearly mention the novelty of your work in a sentence in the introduction based on previous studies.

Response 1: Thank you for this excellent advice. We have now clearly mentioned the novelty of our work in the introduction by highlighting its contribution in relation to previous studies. “Compared with the previous ERB measurements, SR and ER use the identical bolometer-type detectors, and the two are only different in terms of optical design. This configuration minimizes the differences in the instruments themselves which could amplify the system errors by the measurement of the TSI and EOR by two different instruments”.

Point 2: You could also add extra results of the study in a couple of sentences for clarity of (what is exactly you reached to in the study). 

Response 2: In order to enhance the clarity of our study, we have added additional results in a couple of sentences. “The SR and ER are simultaneously sensing the TSI and the terrestrial outgoing radiation, respectively. Furthermore, the SR is installed side by side with three cavities of electrical substitute absolute radiometers, which allows an on-orbit cross-calibration of the SR channel’s absolute level in terms of irradiance. This idea has been verified on a ground-based solar irradiance measurement campaign, the SR has an excellent with the absolute radiometer SIR-2c within 0.3%, which indicates the uncertainty level SR of FY3-F could be one order of magnitude lower the previous PICARD-BOS measurements. “

We believe these revisions have strengthened our manuscript, and we would like to express our gratitude for your input. If you have any further suggestions or comments, please do not hesitate to let us know.

Best regards,

Duo Wu

Reviewer 3 Report

This is a nice work, well explained, with a good description about an instrument for the interesting of many researchers related with climate and solar energy. However there are a few comments that it is neccesary to take into account

There are a lot of acronyms in the article. It would be convenient to put at the beginning a table of nomenclature

Page 2 line 69

Replace “instrument” for “instruments”

Page 3 line 80

Please explain the acronym EOR

Page 3 line 81

ERB and ERM seems to have the same meaning

Page 8, eq(7)

I understand that the black body emission from T1 is much higher than the possible surrounding background radiation, but it is necessary to indicate why this radiation is not considered in the equation

It would be convenient to indicate what are the typical values for the considered shape factors?

page 11

It is necessary to comment on the differences between the SR and the SIAR-2C. Are they significant? What is the origin of these differences?

Page 12

Please provide a comment about the homogeneity of temperature on the surface of extended-area blackbody

Author Response

Dear Reviewer,

Thank you for your positive feedback on our work. We appreciate your recognition of the clarity and relevance of our instrument for researchers in the field of climate and solar energy. We have carefully considered your comments and have addressed them as follows:

Point 1: There are a lot of acronyms in the article. It would be convenient to put at the beginning a table of nomenclature.

Response 1: We agree with your suggestion to include a table of nomenclature at the beginning of the article. This will provide a convenient reference for readers to quickly understand the acronyms used. We have included a comprehensive table of nomenclature to clarify the abbreviations and acronyms employed throughout the manuscript at the end of the article.

Point 2: Page 2 line 69, Replace “instrument” for “instruments”.

Response 2: On page 2, line 69, we have made the correction as recommended, replacing "instrument" with "instruments"

Point 3: Page 3 line 80, Please explain the acronym EOR.

Response 3: On page 3, line 80, we apologize for the oversight in not explaining the acronym "EOR." We have now provided a clear explanation in line 80 for this acronym, indicating its meaning as Earth’s Outgoing Radiation.

Point 4: Page 3 line 81, ERB and ERM seems to have the same meaning.

Response 4: Regarding the concern about ERB and ERM having the same meaning on page 3, line 81, we appreciate your bringing this to our attention. It was indeed an error, ERM means Earth Radiation Measurement which is the name of the instrument on FY-3 series satellite to measure the Earth’s Outgoing Radiation (https://space.oscar.wmo.int/instruments/view/erm_1). we have made the correction and changed the Earth Radiation Budget into Earth Radiation Measurement on line 81.

Point 5: Page 8, eq(7), I understand that the black body emission from T1 is much higher than the possible surrounding background radiation, but it is necessary to indicate why this radiation is not considered in the equation. It would be convenient to indicate what are the typical values for the considered shape factors?

Response 5: In equation (7) on page 8, we understand the need to address the consideration of surrounding background radiation. We have now added a statement to clarify that this equation is use for 180 FOV Spaceborne Relative Radiometer to measure TSI value in space. In this case, the background is the dark space 2.7K temperature. So, the background radiation is not considered in the equation. To provide more context and clarity, we have included typical values for the considered shape factors on page 10, (In the ER condition, the  is about 0.5205.  is about 0.4982) allowing readers to understand the expected range and magnitude of these factors.

Point 6: page 11 It is necessary to comment on the differences between the SR and the SIAR-2C. Are they significant? What is the origin of these differences?

Response 6: we have added a comment comparing the SR and the SIAR-2C in Discussion chapter. Differences between the SR and the SIAR-2C may be due to the different convection effects received by different instruments. The SR and SIAR-2C are simultaneously sensing the TSI and SR have a 1-second cadence however SIAR-2C has a 120-second sampling rate. Furthermore, the SR is installed side by side with three cavities of electrical substitute absolute radiometers, which allows an on-orbit cross-calibration of the SR channel’s absolute level in terms of irradiance. This idea has been verified on a ground-based solar irradiance measurement campaign, the SR has an excellent with the absolute radiometer SIR-2c within 0.3%, which indicates the uncertainty level SR of FY3-F could be one order of magnitude lower the previous PICARD-BOS measurements.

Point 7: Please provide a comment about the homogeneity of temperature on the surface of extended-area blackbody

Response 7: Addressing your comment on page 12, we have included a comment about the homogeneity of temperature on the surface of the extended-area blackbody. According to the test report of the blackbody, the emissivity of the blackbody is 0.9902, and the temperature homogeneity is better than 0.06K.

Reviewer 4 Report

The paper presents the measurement of the Total Solar radiance is well done and well presented.

the paper is recommended for publication.

Author Response

Point 1: The paper presents the measurement of the Total Solar radiance is well done and well presented. the paper is recommended for publication.

Response 1: Thank you for reviewing our manuscript and your positive feedback. We are delighted to hear that you found the measurement of the Total Solar Radiance well done and well presented. We would like to express our gratitude for your time and expertise in evaluating our work. Your positive evaluation encourages us to continue our research in this field.
